# Constrained Dipeptide Surrogates: 5- and 7-Hydroxy Indolizidin-2-one Amino Acid Synthesis from Iodolactonization of Dehydro-2,8-diamino Azelates

**DOI:** 10.3390/molecules27010067

**Published:** 2021-12-23

**Authors:** Ramakotaiah Mulamreddy, William D. Lubell

**Affiliations:** Département de Chimie, Université de Montréal, C.P. 6128 Succursale Centre-Ville, Montreal, QC H3C 3J7, Canada; ramakotaiah.mulamreddy@umontreal.ca

**Keywords:** indolizidin-2-one amino acids, peptide mimic, iodolactonization, lactam, heterocycle

## Abstract

The constrained dipeptide surrogates 5- and 7-hydroxy indolizidin-2-one *N*-(Boc)amino acids have been synthesized from L-serine as a chiral educt. A linear precursor ∆^4^-unsaturated (2*S*,8*S*)-2,8-bis[*N*-(Boc)amino]azelic acid was prepared in five steps from L-serine. Although epoxidation and dihydroxylation pathways gave mixtures of hydroxy indolizidin-2-one diastereomers, iodolactonization of the ∆^4^-azelate stereoselectively delivered a lactone iodide from which separable (5*S*)- and (7*S*)-hydroxy indolizidin-2-one *N*-(Boc)amino esters were synthesized by sequences featuring intramolecular iodide displacement and lactam formation. X-ray analysis of the (7*S*)-hydroxy indolizidin-2-one *N*-(Boc)amino ester indicated that the backbone dihedral angles embedded in the bicyclic ring system resembled those of the central residues of an ideal type II’ β-turn indicating the potential for peptide mimicry.

## 1. Introduction

In peptide science, conformationally constrained dipeptides serve effectively as tools for structure–activity relationship studies to identify biologically active conformers [1,2,3,4,5,6,7,8,9,10,11,12,13,14,15,16,17,18,19,20]. Among approaches for creating constrained dipeptides that employ steric [2,3], stereo-electronic [4,5], and covalent constraints [1,5,6,7,8,9,10,11,12,13,14,15,16,17,18,19,20,21], the use of azabicyclo[X.Y.0]alkanone amino acids offers unique potential for locking the polyamide backbone into specific orientations that may mimic natural secondary structures such as β-turns. Among such bicyclic systems, the azabicyclo[4.3.0]alkanone amino acids, so-called indolizidine-2-one amino acid (I^2^aa) analogs and their ring-substituted derivatives (e.g., **1**–**3**, Figure 1), are among the most studied for utility in dissecting the backbone geometry and side chain alignment responsible for peptide activity towards the development of receptor ligands (e.g., **4**) and enzyme inhibitors (e.g., **5**–**7**) [9,10,11,12,13,14,15,16,17,18,19,20,21].

Several synthetic methods have been developed to introduce substituents at the 5- and 7-positions along the I^2^aa ring system (Figure 1) [9,10,11,12,13,14,15,16,17,18,19]. For example, 5-hydroxy-5-phenyl I^2^aa analogs were synthesized by diastereoselective photochemical cyclization of carbamate-protected β-benzoylalaninyl prolinates [13]. A 5-chloro methyl I^2^aa derivative was synthesized by the treatment of phthalimido allylglycinyl 5-methoxyprolinate with TiCl_4_ in 64% yield [14]. Furthermore, 5-hydroxymethyl, 5-azidomethyl, 5-formyl, 5-carboxy, 5-benzyl, 7-hydroxymethyl, 7-hydroxypropyl, 7-azidopropyl and 7-benzyl, as well as 5,7-dibenzyl I^2^aa derivatives were all synthesized diastereoselectively by routes featuring, respectively, intramolecular displacements and reductive aminations of 4-substituted 5-methanesulfonyl and 5-keto 2,8-diaminoazelates to form 5-substituted prolines, which reacted in lactam cyclization [10,11,12]. Furthermore, 5-iodo I^2^aa diastereomers were respectively prepared by transannular iodolactamization of hexahydro-1H-azonines [15]. Iodide elimination afforded the corresponding ∆^5^-indolizidine-2-one, which was subsequently arylated at the 5-position by oxidative Heck chemistry [16]. In addition, 7-hydroxyethyl, 7-azidoethyl, 7-carboxymethyl, and 7-guanidinylethyl I^2^aas have been synthesized from routes commencing with allylation of glutamic acid [17,22], and utilized in a program towards the development of α_v_β_3_ and α_v_β_5_ integrin receptor ligands [18].

The Hanessian laboratory has played an instrumental role in demonstrating the value of 5- and 7-substituted I^2^aa residues in the study of biologically active peptide receptors [19,20,21]. For example, 5-benzyloxy I^2^aa **4** was designed by Hanessian and shown to be a weak but selective antagonist of the tachykinin NK-2 (neurokinin-2) receptor [19]. Furthermore, 3,5,7-trisubstituted I^2^aas **5**–**7** were designed, synthesized, and shown to act as potent thrombin [Factor IIa] and Factor VIIa inhibitors exhibiting selectivity over plasmin and Factor XIa [20]. Substituted I^2^aa peptides **4**–**7** were respectively synthesized from pyroglutamate by routes featuring the addition of 2-trimethylsilyloxy furan onto an iminium ion intermediate, followed by lactone to lactam ring expansion to obtain the corresponding 5-hydroxy 9-silyloxymethyl indolizidine-2-one [19,20,21]. Subsequent installation of the amine and alkyl substituents at the 3-position and hydroxymethyl group oxidation at the 9-position gave the 3-azido indolizidine-2-one 9-carboxylate counterparts, which were introduced into the peptide mimic structures [19,20,21]. Validating their utility for peptide-based medicinal chemistry, the herculean research of the Hanessian laboratory has illustrated the necessity for effective synthetic routes to access 5- and 7-substituted I^2^aa residues.

Streamlined syntheses of 5- and 7-hydroxy indolizidine-2-one *N*-(Boc)amino acids **2** and **3** are now reported by methods employing L-serine as a chiral educt. Motivated by the research of the Jackson laboratory in which (2*S*,8*S*)-1,9-dibenzyl ∆^4^-2,8-bis[*N*-(Boc)amino]azelate was prepared by the copper-catalyzed S_N_2′ reaction of the zincate derived from *N*-(Boc)-β-iodo alanine benzyl ester onto (*E*)-1,3-dichloroprop-1-ene [23], a series of related ∆^4^-2,8-diaminoazelates were synthesized and studied in different olefin oxidation chemistries to prepare intermediates towards the hydroxy indolizidine-2-one structures. Among different oxidation approaches yielding access to 5-hydroxy and 7-hydroxy I^2^aa derivatives, useful routes to (3*S*,5*S*,6*S*,9*S*)-**2** and (3*S*,6*S*,7*S*,9*S*)-**3** were conceived by way of diastereoselective iodolactonization chemistry inspired by the seminal research of the Bartlett laboratory [24].

## 2. Results and Discussion

Initially, 5- and 7-hydroxy indolizidine-2-one *N*-(Boc)amino esters **8** and **9** were pursued by pathways featuring a ring opening of 4-oxiranyl-2,8-diaminoazelates. Oxiranes **12a**–**c** were synthesized by epoxidation of ∆^4^-2,8-diaminoazelates **11a**–**c**, which were respectively prepared from (*E*)-1,3-dichloroprop-1-ene by copper catalyzed S_N_2′ additions of zincates derived from methyl β-iodo alaninates **12a**–**c** protected with Boc [25], Cbz [26], and Fmoc groups (Figure 1) [27]. Although the 15 Hz coupling constant suggested the formation of the *E*-*trans* olefins **11a** and **11b**, without the corresponding *Z*-*cis* isomer, NOESY experiments were performed to confirm the double-bond geometry. The *E*-geometry of olefins **11a** and **11b** was ascertained by NOESY experiments in which the long-range through-space transfer of magnetization was observed, respectively, between the vinyl C4 (5.38 and 5.35 ppm) and allylic C6 protons (2.09 and 2.07 ppm) and between the vinyl C5 (5.51 and 5.48 ppm) and allylic C3 protons (2.47 and 2.50 ppm) (Figure 1). No nuclear Overhauser effect was observed between the two vinyl protons nor between the two sets of allylic protons.

Previously, epoxidations of *N*-Boc and *N*-Cbz allyl- and homoallyl-glycine esters with *m*-chloroperbenzoic acid (*m*-CPBA) in dichloromethane had given 1:1 diastereomeric mixtures of the corresponding oxiranes, which were inseparable by chromatography [28,29,30]. The C3-protons of benzyl (2*S*,4*RS*)-2-(Boc)amino-3-(2-oxiranyl)propionate was reported to exhibit a doubling of signals in the ^1^H NMR spectrum [28]. The appearance of multiple sets of signals for the two possible isomers was similarly observed in the spectra of inseparable epoxide diastereomers **12a**–**c** and validated by COSY spectra of the Cbz and Fmoc analogs **12b** and **12c** in which through-bond couplings between two sets of C3-protons with two overlapping downfield α-(C2)-proton signals were observed. Oxiranes **12a**–**c** were thus obtained as 1:1 diastereomeric mixtures, which were used in the subsequent chemistry.

Based on the successful synthesis of 6-hydroxymethyl I^2^aa diastereomers in which 5-hydroxymethyl prolines were prepared from a related C2 symmetric oxirane using Lewis-acid activation with BF_3_·Et_2_O in DCM at −78 °C [31], similar conditions were employed for the intramolecular ring-opening of epoxide **12a** (Figure 2). Multiple isomers of the material with a molecular ion corresponding to proline **13** and hydroxyproline **14** were obtained from oxirane **12a** likely by *endo* and *exo* ring openings by the attack of the two different carbamate-protected nitrogen [28,32,33]. Considering that the isomeric mix could be due, in part, to carbocation intermediates formed under the Lewis acid conditions, a method to remove the Boc group without the ring opening of the epoxide was attempted featuring heating oxirane **12a** in water at reflux [34]. Deprotection of the Boc group, intramolecular epoxide ring opening, and lactam formation all occurred upon treating **12a** with boiling water. Amine protection with di-*tert*-butyl dicarbonate and triethyl amine in dichloromethane, however, afforded four isomers of 5- and 7-hydroxy I^2^aa esters **8** and **9**, which were observed by LCMS in a 1:1:1:1 ratio. Employing Cbz-protected epoxide **12b**, hydrogenolytic cleavage of the carbamate using hydrogen and palladium-on-carbon in ethanol commenced an epoxide ring opening and lactam formation sequence, which was followed by Boc protection as described above to afford four isomers of **8** and **9**, which were observed in a 1:5:5:1 ratio by HPLC. The improvement in selectivity may be due to a favored *exo*-*tet*-like ring opening of the epoxide diastereomers by the free amine, which when generated at a lower temperature reacted to favor the proline instead of the hydroxyproline counterparts [32,33]. In spite the possibility of improved regioselectivity in the oxirane ring opening, the route (Figure 2) was, however, deemed inefficient due to the complications engendered from the lack of diastereomeric selectivity in the epoxidation of olefins **11**.

Prompted by earlier success using transannular iodolactamization to prepare azabicyclo[X.Y.0]alkan-2-one ring systems [15,35], and related iodoamination protocols for preparing iodomethyl pyrrolidines and piperidines [36,37,38], ∆^4^-diaminoazelate **11a** was subjected to iodine and NaHCO_3_ at −20 °C (Figure 3). The ring opening of the iodonium intermediate by one of the two carbamate-protected nitrogen appeared to be a method for selectively obtaining proline **15** instead of the azetidine counterpart; however, a mixture of diastereomeric iodolactones **16** was also produced as a competing side product. Considering the lactone as a potential means for differentiating between the two carboxylates, dihydroxylation of ∆^4^-diaminoazelate **11a** was performed using osmium tetroxide and *N*-methylmorpholine *N*-oxide (NMO) in aqueous acetone to provide hydroxy lactone **17** as a mixture of diasteromers [39]. Mesylate **18** was obtained by methanesulfonation of hydroxy lactone **17** using methanesulfonyl chloride and triethylamine in dichloromethane. Mesylate **18** was converted to hydroxy I^2^aa analogs **8** and **9** by a three-step sequence featuring proline formation after Boc group removal with HCl gas bubbles in dichloromethane, lactam cyclization upon treatment of the hydrochloride salt with triethylamine in methanol at reflux, and amine protection with di-*tert*-butyl decarbonate in dichloromethane. The HPLC chromatogram of the products from this sequence exhibited four peaks with molecular ions corresponding to 5- and 7-hydroxy Boc-I^2^aa-OMe isomers **8** and **9** (Figure 3) in a 1:1:1:1 ratio.

Different mixtures of 5- and 7-hydroxy Boc-I^2^aa-OMe diastereomers **8** and **9** likely arose from a combination of a lack of facial selectivity in the epoxidation and the dihydroxylation of olefin **11** and competing nucleophilic attack from both nitrogen of diamino azelate epoxide **12** and methanesulfonate **18**. The loss of stereochemical integrity may also arise from competing S_N_1 processes due to the epoxide ring opening prior to pyrrolidine formation. Intrigued by the production of iodolactone **16** as a side product from the iodoamination strategy, an iodolactonization approach was considered because of the high facial selectivity achieved on simpler γ,δ-unsaturated carboxylic acids [24,40,41].

After saponification of diester **11a** with lithium hydroxide in aqueous dioxane, dicarboxylic acid **19** was treated with cesium carbonate and iodine in an ice-cold acetonitrile solution (Figure 4). Analysis by LCMS demonstrated a major peak with a molecular ion corresponding to lactone **20**. Subsequent treatment with iodomethane and potassium carbonate in DMF furnished the corresponding methyl ester tetrahydrofuran-2-one (1′*R*,5*S*)-**16** after chromatography in 55% yield from diacid acid **19**. Attempts to perform the iodolactonization without a base gave a product mostly from the loss of Boc protection. Employing the same three-step sequence described above to convert methane sulfonate **18** into esters **8** and **9**, iodide (1′*R*,5*S*)-**16** was transformed into separable 5- and 7-hydroxy I^2^aa esters (5*S*,6*S*)-**8** and (6*S*,7*S*)-**9** in 42% and 34% overall yields, respectively. Subsequent saponification of esters (5*S*,6*S*)-**8** and (6*S*,7*S*)-**9** gave, respectively, the acids (5*S*,6*S*)-**2** and (6*S*,7*S*)-**3** in 64% and 78% yields.

### Assignment of Regio-Chemistry and Stereochemistry of 5- and 7-Hydroxy I^2^aa Esters

The configuration of the ring fusion and hydroxyl group carbons of the 5- and 7-hydroxy I^2^aa esters **8** and **9**, as well as the alcohol position on the ring system, were all assigned based on two-dimensional NMR spectroscopic experiments. The locations of the indolizidine-2-one ring protons were initially assigned by COSY experiments in which through-bond couplings were used to trace the sequence from the downfield shifted carbamate NH to the C9 hydrogen. Subsequently, heteronuclear single quantum coherence (HSQC) spectroscopy was used to correlate the protons linked to similar carbons. The β-protons on the same face as the C3 carbamate and C9 carboxylate appeared generally up-field of their α-counterparts due to anisotropic effects caused by the latter functional groups [42]. Finally, relative configurations were ascertained (Figure 2) based on NOESY experiments in which the observed through-space transfers of magnetization were used to correlate the stereochemical assignments.

The ring fusion protons (3.88 and 3.74 ppm) of 5- and 7-hydroxy Boc-I^2^aa-OMe (5*S*,6*S*)-**8** and (6*S*,7*S*)-**9** were respectively assigned the *S* stereochemistry based on nuclear Overhauser effects (nOe) with the C4β and C8β protons (1.99 and 1.84 ppm) and with the C3 proton (4.13 ppm, Figure 2). No long-range through-space transfer of magnetization was observed for the protons on the alcohol-bearing carbons. In the case of (6*S*,7*S*)-**9**, the relative nOe between the C7 proton was stronger for the C8α proton (2.35 ppm) compared to that of the C8β proton (2.15 ppm). The stereochemical assignments for Boc-(7-OH)I^2^aa-OMe (6*S*,7*S*)-**9** were confirmed by X-ray analysis as discussed below.

The configurations of the hydroxyl group in Boc-(5-OH)I^2^aa-OMe (5*S*,6*S*)-**8** and the iodolactone of tetrahydrofuran-2-one (1′*R*,5*S*)-**16** were based on the latter serving as a common intermediate for both the former and Boc-(7-OH)I^2^aa-OMe (6*S*,7*S*)-**9**. The stereochemistry of the ring-fusion and alcohol carbons are respectively derived from the inversion on nitrogen attack of the iodide and retention on the lactone opening during synthesis of the bicycle. Although the order of attack of the iodine and carboxylate may proceed by a traditional iodonium intermediate (Figure 4) [24], and by a more concerted nucleophile-assisted alkene activation mechanism [43], the stereochemical outcome of iodolactone (1′*R*,5*S*)-**20** arises from the attack of iodine by the face of the olefin on the opposite side of the proximal carboxylate of ∆^4^-azelate **19** (Figure 4).

The relative configurational assignments for 7-hydroxy Boc-I^2^aa-OMe (6*S*,7*S*)-**9** were confirmed by X-ray analysis of crystals grown from a dichloromethane-in-hexanes mixture (Figure 3). Two conformers differing primarily by the carbamate orientation were present in the unit cell and connected by an intermolecular hydrogen bond from the 7-hydroxyl group donor to the lactam carbonyl oxygen acceptor. Examination of the backbone dihedral angles embedded in the I^2^aa ring system (*ψ*^i+1^ −172° and *ϕ^i^*^+2^ −78°; *ψ*^i+1^ −175° and *ϕ*^i+2^ −71°) of the conformers in the X-ray structure of the 7-hydroxy analog (6*S*,7*S*)-**9** indicated a close relation to those of the central residues of an ideal type II’ β-turn (*ψ*^i+1^ –120° and *ϕ*^i+2^ −80°) [44], and to that of the methyl ester of the parent I^2^aa counterpart (6*S*)-**21** (*ψ*^i+1^ −176° and *ϕ*^i+2^ −78°, Figure 4) [45]. Relative to the values in the crystal structure of Boc-I^2^aa-OMe (6*S*)-**21**, the *ϕ*^i+2^ dihedral angle was apparently less influenced by the smaller 7β-hydroxy substituent than the 7α-hydroxymethyl substituent in Boc-(7-HOCH_2_)I^2^aa-OMe (**22**, *ψ*^i+1^ −175° and *ϕ*^i+2^ −68°) [11].

## 3. Materials and Methods

Anhydrous solvents (CH_3_CN, DMF, (CH_3_)_2_CO, CH_2_Cl_2_, and CH_3_OH) were obtained by passage through solvent filtration systems (GlassContour, Irvine, CA, USA). All reagents from commercial sources were used as received: Iodine was purchased from Aldrich (USA) and solvents were obtained from Fisher Chemical. The *N*-(Boc)-, (Cbz)-, and (Fmoc)-3-iodo-l-alanine methyl esters **10a**–**c** were respectively prepared according to the literature methods reported in references [25,26,27]. Purification by silica gel chromatography was performed on 230−400 mesh silica gel; analytical thin-layer chromatography (TLC) was performed on silica gel 60 F254 (aluminum sheet) and visualized by UV absorbance or staining with KMnO_4_. Melting points are reported in degree Celsius (°C), uncorrected and obtained using a Mel-Temp melting point apparatus equipped with a thermometer on the sample that was placed in a capillary tube. Spectroscopic ^1^H and ^13^C NMR experiments were recorded at room temperature (298 K) in CDCl_3_ (7.26/77.16 ppm), DMSO-*d*_6_ (2.5/39.56), and CD_3_OD (3.31/49.0 ppm) on Bruker AV (500/125, and 700/175 MHz) instruments using an internal solvent as the reference. Spectra are presented in the Appendix A. Chemical shifts are reported in parts per million (ppm), and coupling constant (*J*) values in Hertz (Hz). Abbreviations for peak multiplicities are s (singlet), d (doublet), t (triplet), q (quadruplet), q (quintuplet), m (multiplet), and br (broad). Certain ^13^C NMR chemical shift values were extracted from HSQC spectra. High-resolution mass spectrometry (HRMS) data were obtained on an LC-MSD instrument in electrospray ionization (ESI-TOF) mode by the Centre Régional de Spectrométrie de Masse de l’Université de Montréal. Either protonated molecular ions [M + H]^+^ or sodium adducts [M + Na]^+^ were used for empirical formula confirmation. Infrared spectra were recorded in the neat on a Perkin Elmer Spectrometer FT-IR instrument, and are reported in reciprocal centimeters (cm^−1^). The X-ray structure was solved using a Bruker Venture Metaljet diffractometer by the Laboratoire de diffraction des rayons X de l’Université de Montréal. Specific rotations [α]_D_ were measured at 25 °C at the specified concentrations (*c* in g/100 mL) using a 0.5 dm cell on a PerkinElmer Polarimeter 589 instrument and expressed using the general formula [α]_D_^25^ = (100 × α)/(d × *c*).

### 3.1. (3S,5S,6S,9S)-3-N-(Boc)amino-5-hydroxy-indolizin-2-one-9-carboxylic Acid [(3S,5S,6S,9S)-***2***]

A 0 °C solution of ester (3*S*,5*S*,6*S*,9*S*)-**8** (15 mg, 0.046 mmol) in 1,4-dioxane (0.5 mL) was treated with a 1N solution of LiOH (1.9 mg, 0.046 mmol, 1 equiv.). The cooling bath was removed. The reaction mixture was warmed to room temperature with stirring overnight, at which time TLC indicated the consumption of the starting material. The volatiles were evaporated under reduced pressure. The residue was partitioned between H_2_O (5 mL) and ethyl acetate (5 mL). The aqueous phase was acidified with 1 N HCl to pH 3 and extracted with ethyl acetate (3 × 10 mL). The organic extractions were combined, dried with Na_2_SO_4_, filtered, and concentrated under vacuum to afford (3*S*,5*S*,6*S*,9*S*)-**2** (9 mg, 64%) as a white foam; [α]_D_^25^ –10.2 (*c* 0.32, CHCl_3_); ^1^H NMR (500 MHz, CDCl_3_): δ 5.39 (s, br, 1H), 4.71 (s, 1H), 4.39 (s, br, 1H), 4.29–4.28 (m, 1H), 3.84–3.80 (m, 1H), 2.6–2.52 (m, 1H), 2.39-2.33 (m, 2H), 2.26-2.20 (m, 1H), 2.05-2.02 (m, 1H), 2.0-1.95 (m, 1H), 1.67-1.63 (m, 1H), 1.47 (s, 9H); ^13^C{^1^H} NMR (125 MHz, CDCl_3_) δ 172.0, 167.3, 147.3, 80.5, 64.0, 60.0, 35.2, 32.0, 30.0, 28.3, 26.1, 23.0; FT-IR (neat) ν_max_ 3328, 2919, 1702, 1521, 1449, 1362, 1208, 1166, 1050, 1031 cm^−1^; HRMS (ESI-TOF) *m*/*z* [M + Na]^+^ calcd for C_14_H_22_N_2_O_6_Na 337.1370, found 337.1374.



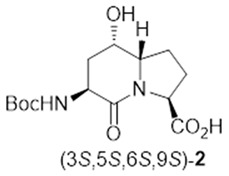



### 3.2. (3S,6S,7S,9S)-3-N-(Boc)amino-7-hydroxy-indolizin-2-one-9-carboxylic Acid [(3S,6S,7S,9S)-***3***]

A 0 °C solution of ester (3*S*,6*S*,7*S*,9*S*)-**9** (150 mg, 0.46 mmol) in 1,4-dioxane (5 mL) was treated with a 1N solution of LiOH (19.2 mg, 0.46 mmol, 1 equiv.). The cooling bath was removed. The reaction mixture was warmed to room temperature with stirring for 3 h, at which time TLC indicated the consumption of the starting material. The volatiles were evaporated under reduced pressure. The residue was partitioned between H_2_O (10 mL) and ethyl acetate (5 mL). The aqueous phase was acidified with 1 N HCl to pH 3 and extracted with ethyl acetate (3 × 10 mL). The organic extractions were combined, dried with Na_2_SO_4_, filtered, and concentrated under vacuum to afford (3*S*,6*S*,7*S*,9*S*)-**3** (112 mg, 78%) as a white solid: mp 105–106 °C; [α]_D_^25^ –19.13 (*c* 0.23, CHCl_3_); ^1^H NMR (500 MHz, CD_3_OD): δ 4.460–4.43 (dd, *J* = 9.3, 4.3 Hz, 1H), 4.26–4.24 (m, 1H), 4.22–4.17 (m, 1H), 3.76–3.72 (m, 1H), 2.47–2.41 (m, 1H), 2.21–2.18 (d, *J* = 14.2 Hz, 1H), 2.15–2.07 (m, 2H), 1.87–1.82 (m, 2H), 1.48 (s, 9H); ^13^C{^1^H} NMR (125 MHz, CD_3_OD) δ 174.0, 170.0, 156.6, 79.1, 71.2, 62.3, 57.2, 50.0, 37.0, 27.3, 27.0, 19.0; FT-IR (neat) ν_max_ 3325, 2922, 1697, 1523, 1451, 1365, 1211, 1162, 1055, 1032 cm^−1^; HRMS (ESI-TOF) *m*/*z* [M + Na]^+^ calcd for C_14_H_22_N_2_O_6_Na 337.1370, found 337.1374.



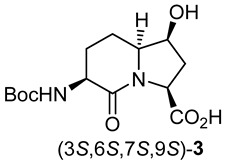



### 3.3. Methyl (3S,5S,6S,9S)-3-N-(Boc)amino)-5-hydroxy-indolizin-2-one-9-carboxylate and (3S,6S,7S,9S)-3-N-(Boc)amino)-7-hydroxy-indolizin-2-one-9-carboxylate [(3S,5S,6S,9S)-***8*** and (3S,6S,7S,9S)-***9***]

A solution of (1′*R*,5*S*)-1′-iodo-tetrahydrofuran-2-one (1′*R*,5*S*)-**16** (1.0 g, 1.8 mmol) in dichloromethane (20 mL) was treated with HCl gas bubbles for 2-3 h, when TLC indicted complete consumption of the starting carbamate and LCMS analysis indicated a new peak RT = 0.7 min (C18 column, 10:90 CH_3_CN:H_2_O) with a molecular ion of [M + H]^+^ *m*/*z* 357. The reaction mixture was evaporated to a residue, which was dissolved in MeOH (5 mL), treated with triethylamine (545 mg, 5.4 mmol, 3 equiv.), and heated at reflux using an oil bath overnight, when LCMS indicated a new peak RT = 0.68 min (eluent C18 column, 10:90 CH_3_CN:H_2_O) with the molecular ion [M + H]^+^ *m*/*z*. The volatiles were evaporated under reduced pressure. The residue was dissolved in dichloromethane (10 mL), treated with (Boc)_2_O (0.14 g, 0.63 mmol, 1.2 equiv.), and stirred for 3 h, when TLC indicated two new spots and LCMS indicated a new peak RT = 5.0 min (C18 column, 10:90 CH_3_CN:H_2_O). The volatiles were removed under reduced pressure. The residue was purified by flash column chromatography using 60–80% EtOAc in hexanes as eluent.



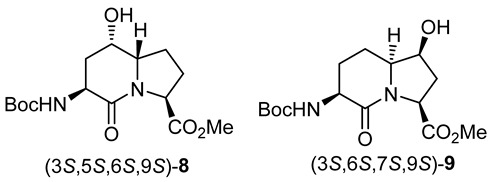



The first to elute was Boc-(7-HO)I^2^aa-OMe (3*S*,6*S*,7*S*,9*S*)**-9** (200 mg, 34%) as a white solid: mp 138–140 °C; R*_f_* = 0.47, (100% EtOAc twice eluted, visualized with KMnO_4_); [α]_D_^25^ −28.2 (*c* 0.85, CHCl_3_); ^1^H NMR (500 MHz, CDCl_3_) δ 5.14 (s, br, NH), 4.46–4.44 (dd, *J* = 10 Hz, 1H), 4.22–4.18 (m, 1H), 4.15–4.12 (m, 1H), 3.84 (s, 3H), 3.76–3.71 (m, 1H), 3.61–3.58 (d, *J* = 15 Hz, OH), 2.4–2.33 (m, 2H), 2.29–2.22 (m, 1H), 2.15–2.12 (dt, *J* = 14.5, 0.9 Hz, 1H), 2.0–1.93 (m, 1H), 1.77–1.69 (m, 1H), 1.47 (s, 9H); ^13^C{^1^H} NMR (125 MHz, CDCl_3_) δ 175.5, 170.1, 156.0, 80.0, 73.0, 61.5, 57.1, 53.2, 51.0, 36.3, 28.3, 27.1, 19.2; FT-IR (neat) ν_max_ 3357, 2979, 1693, 1636, 1518, 1437, 1392, 1365, 1249, 1165, 1099, 1063, 1005 cm^−1^ HRMS (ESI-TOF) *m*/*z* [M + Na]^+^ calcd for C_15_H_24_N_2_O_6_Na 351.1526 found 351.1522.

Next to elute was Boc-(5-HO)I^2^aa-OMe (3*S*,5*S*,6*S*,9*S*)-**8** (250 mg, 42%) as a white solid: mp 75–77 °C; R*_f_* = 0.3 (100% EtOAc, twice eluted, visualized with KMnO_4_); [α]_D_^25^ –12.6 (*c* 0.75, CHCl_3_). ^1^H NMR (500 MHz, CDCl_3_) δ 5.25 (s, br, NH), 4.50–4.44 (m, 2H), 4.27 (s, 1H), 3.89-3.86 (m, 1H), 3.77 (s, 3H), 2.74–2.68 (m, 1H), 2.44–2.40 (m, 1H), 2.11–2.04 (m, 2H), 2.00–1.97 (m, 1H), 1.95–1.92 (m, 1H), 1.88–1.83 (m, 1H), 1.45 (s, 9H); ^13^C{^1^H} NMR (125 MHz, CDCl_3_) δ 173.0, 168.0, 156.2, 80.0, 64.0, 63.3, 58.2, 52.3, 47.4, 36.0, 28.3, 28.0, 27.0; FT-IR (neat) ν_max_ 3360, 2983, 1702, 1633, 1518, 1438, 1395, 1250, 1162, 1102, 1002 cm^−1^; HRMS (ESI-TOF) *m*/*z* [M + Na]^+^ calcd for C_15_H_24_N_2_O_6_Na 351.1526 found 351.1522.

### 3.4. Dimethyl (2S,4E,8S)-∆^4^-2,8-(di-N-(Boc)amino)azelate (***11a***)

In a 250-mL round bottom flask, fitted with a three-way stopcock, CuBr•DMS (1.22 g, 0.006 mol, 0.13 equiv.) was weighed, dried gently with a heat gun under vacuum until the powder changed color from white to light green, placed under argon, treated with dry DMF (30 mL), followed by (*E*)-1,3-dichloroprop-1-ene (2.5 g, 0.023 mol, 0.5 equiv.). In a Schlenk tube, zinc (8.9 g, 0.14 mol, 3 equiv.) and iodine (0.35 g, 0.0014 mol, 0.03 equiv.) were mixed under an argon atmosphere, and thrice heated under vacuum with a heat gun for 10 min and cooled under a flush of argon. A solution of *N*-(Boc)-3-iodo-l-alanine methyl ester **10a** (15 g, 0.046 mol) in dry DMF (30 mL) was added to the Schlenk tube and stirred for 1h, when TLC analysis confirmed the consumption of the iodide (R*_f_* = 0.7, 30% EtOAc in hexanes) and formation of the organozinc reagent (R*_f_* = 0.2, 30% EtOAc in hexanes). Stirring was stopped, the excess zinc powder was allowed to settle, and the supernatant was transferred dropwise via a syringe with care to minimize the transfer of zinc into the flask containing the copper catalyst. After stirring at rt overnight, TLC indicated a new spot (R*_f_* = 0.48, 40% EtOAc in hexanes) and the reaction mixture was diluted with ethyl acetate (150 mL), stirred for 15 min, and filtered through a silica gel pad. The filtrate was treated with water (100 mL), transferred into a separatory funnel, and diluted with ethyl acetate (50 mL). The organic phase was washed successively with 1 M Na_2_S_2_O_3_ (2 × 100 mL), water (4 × 100 mL), and brine (2 × 100 mL), dried over Na_2_SO_4_, filtered, and evaporated. The volatiles were removed under reduced pressure to afford a residue that was purified by chromatography using 25–30% EtOAc in hexanes as the eluent. Evaporation of the collected fractions gave azelate **11a** (11.4 g, 56%) as a colorless liquid: R*_f_* = 0.48 (2:3 EtOAc/Hexanes, visualized with KMnO_4_); [α]_D_^25^ +25.2 (*c* 1.04, CHCl_3_); ^1^H NMR (500 MHz, CDCl_3_) δ 5.54–5.48 (dt, *J* = 15, 5 Hz, 1H), 5.39–5.34 (dt, *J* = 15, 5 Hz, 1H), 5.25–5.24 (d, *J* = 5.0 Hz, 1H), 5.03–5.01 (d, *J* = 10 Hz, 1H), 4.40–4.37 (m, 1H), 4.34–4.30 (m, 1H), 3.76 (s, 3H), 3.75 (s, 3H), 2.52–2.43 (m, 2H), 2.12–2.07 (m, 2H), 1.90–1.84 (m, 1H), 1.71–1.67 (m, 1H), 1.47 (s, 9H), 1.46 (s, 9H); ^13^C{^1^H} (125 MHz, CDCl_3_) δ 173.3, 173.0, 155.3, 155.2, 133.1, 125.5, 79.95, 79.84, 53.2, 53.0, 52.3, 52.2, 35.6, 32.4, 28.4, 28.3, 23.2; FT-IR (neat) ν_max_ 3363, 2977, 1698, 1508, 1437, 1391, 1365, 1247, 1211, 1157, 1103, 1050, 1021 cm^−1^; HRMS (ESI-TOF) *m*/*z* [M + Na]^+^ calcd for C_21_H_36_N_2_O_8_Na 467.2363 found 467.2359.



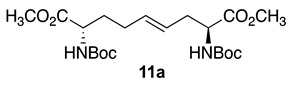



### 3.5. Dimethyl (2S,4E,8S)-∆^4^-2,8-(di-N-(Cbz)amino)azelate (***11b***)

Diamino azelate **11b** with Cbz protection was synthesized according to the protocol described for the synthesis of Boc counterpart **11a** using *N*-(Cbz)-3-iodo-l-alanine methyl ester **10b** (8.0 g, 0.02 mmol) and isolated as a colorless liquid (3.5 g, 63%): R*_f_* = 0.30 (2:3 E.A/Hexanes, visualized by UV); [α]_D_^25^ +15.9 (c 1.09, CHCl_3_); ^1^H NMR (500 MHz, CD_3_OD): δ 7.40–7.31(m, 10H), 5.57–5.55 (d, *J* = 10 Hz, 1H), 5.52–5.45 (dt, *J* = 15, 5 Hz, 1H), 5.38–5.33 (dt, *J* = 15, 5 Hz, 1H), 5.29–5.27 (d, *J* = 10Hz, 1H), 5.16–5.11(m, 4H), 4.48–4.44 (m, 1H), 4.42–4.37 (m, 1H), 3.76 (s, 3H), 3.75 (s, 3H) 2.58–2.46 (m, 2H), 2.13–2.01 (m 2H), 1.94–1.82 (m, 1H), 1.74–1.67 (m, 1H); ^13^C{^1^H} NMR (125 MHz, CDCl_3_) δ 172.2, 156.0, 136.2, 132.3, 128.57, 128.54, 128.52, 128.46, 128.25, 128.22, 128.16, 128.13, 125.32, 67.1, 67.0, 54.0, 53.0, 52.4, 52.3, 35.4, 32.4, 32.2, 28.2; FT-IR (neat) ν_max_ 3332, 2953, 1699, 1521, 1437, 1341, 1207, 1050 cm^−1^; HRMS (ESI-TOF) *m*/*z* [M + H]^+^ calcd for C_27_H_33_N_2_O_8_ 513.2231, found 513.2234.



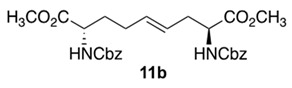



### 3.6. Dimethyl (2S,4RS,5RS,8S)-2,8-di-N-(Boc)amino-4-oxiranyl-azelate (***12a***)

A solution of ∆^4^-di-*N*-(Boc)aminoazelate **11a** (2.0 g, 4.5 mmol) in dichloromethane (DCM, 30 mL) was cooled to 0 °C and treated with *m*-chloroperoxybenzoic acid (2.0 g, 9.0 mmol, 2.0 equiv.). The ice bath was removed. The suspension was warmed to room temperature with stirring overnight, when TLC showed the complete consumption of olefin **11a** (R*_f_* = 0.48, 40% EtOAc in hexanes) and a new polar spot for epoxide **12a** (R*_f_* = 0.2, 40% EtOAc in hexanes). The reaction mixture was diluted with DCM (30 mL), transferred to a separatory funnel, and washed sequentially with 1N NaOH (2 × 20 mL), water (20 mL), and brine (20 mL), dried over Na_2_SO_4_, filtered, and concentrated under vacuum to a residue that was purified by flash column chromatography using 20% EtOAc in hexanes as the eluent. Evaporation of the collected fractions afforded epoxide **20a** (1.75 g, 84%) as colorless oil: R*_f_* = 0.2 (2:3 EtOAc/hexanes, visualized with KMnO_4_); [α]_D_^25^ +2.5 (*c* 0.81, CHCl_3_); ^1^H NMR (500 MHz, CD_3_OD): δ 4.32–4.26 (m, 1H), 4.18–4.13 (m, 1H), 3.75 (s, 3H), 3.74 (s, 3H), 2.87–2.75 (m, 2H), 1.97–1.90 (m, 2H), 1.80–1.72 (m, 1H), 1.64–1.59 (m, 1H), 1.47 (s, 20H); ^13^C{^1^H} NMR (125 MHz, CD_3_OD) δ 173.2, 173.0, 156.7, 156.6, 79.4, 79.2, 58.0, 57.5, 55.4, 55.3, 53.5, 53.1, 51.51, 51.4, 51.3, 34.0, 27.3; FT-IR (neat) ν_max_ 3326, 2955, 1699, 1523, 1437, 1210, 1045, 912 cm^−1^; HRMS (ESI-TOF) *m*/*z* [M + Na]^+^ calcd for C_21_H_36_N_2_O_9_Na 483.2313, found 483.2321.



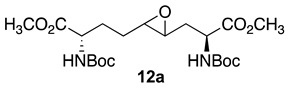



### 3.7. Dimethyl (2S,4RS,5RS,8S)-2,8-di-N-(Cbz)amino-4-oxiranyl-azelate (***12b***)

Epoxide **12b** with Cbz protection was synthesized using the protocol described for the preparation of Boc counterpart **12a** using dimethyl ∆^4^-di-(Cbz)amino azelate **11b** (3.2 g, 6.2 mmol) and isolated as a colorless liquid (2.5g, 76%): R*_f_* = 0.21 (2:3 EtOAc/hexanes, visualized by UV); [α]_D_^25^ +7.95 (c 0.88, CHCl_3_); ^1^H NMR (500 MHz, CDCl_3_): δ 7.40–7.33 (m, 10H), 5.68–5.62 (d, *J* = 10Hz, 1H), 5.44–5.32 (d, *J* = 5Hz, 1H), 5.16–5.11 (m, 4H), 4.58–4.11 (m, 1H), 4.45–4.39 (s, 1H), 3.79–3.76 (s, 6H), 2.81–2.70 (m, 2H), 2.25–2.07 (m, 1H), 2.04–1.92 (m, 2H), 1.83–1.75 (m, 1H), 1.71–1.65 (m, 1H), 1.54–1.44 (m, 1H); ^13^C{^1^H} NMR (125 MHz, CDCl_3_) δ 172.5, 172.1, 156.0, 136.2, 128.6, 128.3, 128.2, 67.1, 57.5, 55.3, 55.1, 53.5, 53.2, 53.0, 52.65, 52.57, 52.51, 52.2, 35.0, 30.0, 29.0, 28.0, 27.5; FT-IR (neat) ν_max_ 3332, 2953, 1700, 1521, 1437, 1344, 1208, 1049 cm^−1^; HRMS (ESI-TOF) *m*/*z* [M + H]^+^ calcd for C_27_H_33_N_2_O_9_ 529.2180, found 529.2190.



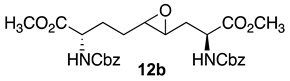



### 3.8. Dimethyl (2S,4RS,5RS,8S)-2,8-di-N-(Fmoc)amino-4-oxiranyl-azelate (***12c***)

Dimethyl (2*S*,4*E*,8*S*)-∆^4^-2,8-(di-*N*-(Fmoc)amino)azelate (**11c**) was synthesized using the protocol described for the synthesis of ∆^4^-di-(Boc)amino azelate **11a** from *N*-(Fmoc)-3-iodo-L-alanine methyl ester (**10c**, 1.5 g, 0.0022 mol) and isolated as a colorless liquid (0.7 g, 63%): R*_f_* = 0.21 (4:6 ethyl acetate/hexanes, visualized by UV). Epoxidation was performed as described for Boc counterpart **11a** using dimethyl (2*S*,4*E*,8*S*)-∆^4^-2,8-(di-*N*-(Fmoc)amino)azelate (**11c**, 600 mg, 0.87 mmol), which gave a colorless solid (500 mg, 82%): mp 89–92 °C; R*_f_* = 0.30 (4:6 EtOAc/hexanes, visualized by UV); [α]_D_^25^ +5.5 (*c* 0.51, CHCl_3_); ^1^H NMR (500 MHz, CDCl_3_): δ 7.79–7.77 (d, *J* = 10 Hz, 4H) 7.63–7.57 (m, 4H), 7.43–7,40 (m, 4H), 7.34–7.31 (m, 4H), 5.74–5.67 (dd, *J* = 10, 5 Hz, 1H), 5.48–5.34 (dd, *J* = 12, 10 Hz, 1H), 4.60–4.51 (m, 2H), 4.46–4.40 (m, 4H), 4.26–4.22 (m, 2H), 3.81 (s, 3H), 3.78 (s, 3H), 2.85–2.73 (m, 2H), 2.16–1.74 (m, 6H); ^13^C{^1^H} NMR (125 MHz, CDCl_3_) δ 172.6, 172.1, 156.0, 143.8, 143.7, 141.3, 130.0, 128.0, 127.1, 125.1, 120.0, 67.2, 67.1, 67.0, 57.4, 55.3, 55.1, 53.2, 52.73, 52.7, 52.6, 52.5, 47.1, 35.0, 30.0, 28.97, 28.9, 27.6, 27.5; FT-IR (neat) ν_max_ 3290, 2952, 1690, 1531, 1448, 1260, 1215, 1085, 1045 cm^−1^; HRMS (ESI-TOF) *m*/*z* [M + H]^+^ calcd for C_41_H_41_N_2_O_9_ 705.2806, found 705.2819.



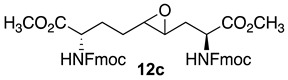



### 3.9. (2. S,4E,8S)-∆^4^-2,8-(di-N-(Boc)amino)azelic Acid (***19***)

A 0 °C solution of dimethyl (2*S*,4*E*,8*S*)-∆^4^-2,8-(di-*N*-(Boc)amino)azelate **(11a**, 500 mg, 1.12 mmol) in 1,4-dioxane (5 mL) was treated with a 1N solution of LiOH (94.4 mg, 2.25 mmol, 2 equiv.). The cooling bath was removed. The reaction mixture was warmed to room temperature with stirring for 3 h, at which time TLC indicated the consumption of the starting material. The volatiles were evaporated under reduced pressure. The residue was partitioned between H_2_O (10 mL) and EtOAc (5 mL). The aqueous phase was acidified with 1 N HCl to pH 3 and extracted with ethyl acetate (3 × 10 mL). The organic extractions were combined, dried with Na_2_SO_4_, filtered, and concentrated under vacuum to afford diacid **19** (430 mg, 92%) as a white solid: mp 71–73 °C; [α]_D_^25^ +39.0 (*c* 0.82, CHCl_3_); ^1^H NMR (500 MHz, DMSO-*d*_6_): δ 12.42 (s, 2H), 7.08–7.07 (d, *J* = 5.0 Hz, 1H), 6.97–6.96 (d, *J* = 5 Hz, 1H), 5.50–5.44 (m, 1H), 5.40–5.35 (m, 1H), 3.90–3.84 (m, 2H), 2.37–2.32 (m, 1H), 2.29–2.23 (m, 1H), 1.71–1.50 (m, 4H), 1.39 (s, 9H), 1.38 (s, 9H); ^13^C{^1^H} NMR (125 MHz, DMSO-*d*_6_) δ 175.0, 174.0, 156.03, 155.88, 132.2, 127.0, 78.47, 78.41, 60.2, 54.1, 53.3, 34.5, 31.1, 28.68, 28.66; FT-IR (neat) ν_max_ 3697, 2980, 1694, 1507, 1393, 1367, 1245, 1157, 1053, 1033, 1018 cm^−1^; HRMS (ESI-TOF) *m*/*z* [M + Na]^+^ calcd for C_19_H_32_N_2_O_8_Na 439.2050, found 439.2070.



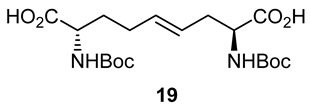



### 3.10. (1′R,5S)-3-N-(Boc)amino-5-[1′-iodo-4′-N-(Boc)amino-4′-methoxcarbonylbutyl]-tetrahydrofuran-2-one [(1′R,5S)-***16***]

A 0 °C mixture of carboxylic acid (1′*R*,5*S*)-**20** (2.1 g, 3.87 mmol) and K_2_CO_3_ (800 mg, 5.8 mmol, 1.5 equiv.) in DMF (20 mL) was treated with methyl iodide (820 mg, 5.8 mmol, 1.5 equiv.). The ice bath was removed. After stirring for 2–3 h, the room temperature mixture exhibited a nonpolar spot (2:3 EtOAc/hexanes) by TLC and indicated a new peak at RT = 9.0 min (C18 column, 10:90 CH_3_CN:H_2_O) by LCMS analysis, with a molecular ion of [M + Na]^+^ *m*/*z* 579. The reaction mixture was diluted with water and extracted with ethyl acetate (4 × 50 mL). The ethyl acetate layer was washed with water (4 × 50 mL) and brine (2 × 30 mL), dried over Na_2_SO_4_, filtered, and concentrated under reduced pressure. The residue was purified by flash column chromatography using 20–30% EtOAc in hexanes as the eluent. Evaporation of the collected fractions gave tetrahydrofuran-2-one (1′*R*,5*S*)-**16** (1.1g, 55% from diacid **19**) as a colorless solid: mp 58–60 °C; R*_f_* = 0.56 (2:3 EtOAc/hexanes, visualized by KMnO_4_), [α]_D_^25^ +13.4 (*c* 0.68, CHCl_3_); ^1^H NMR (500 MHz, CDCl_3_): δ 5.10–5.08 (d, *J* = 10Hz, 2H), 4.45–4.41 (m, 1H), 4.37–4.33 (m, 2H), 4.08–4.04 (t, *J* = 10 Hz, 1H), 3.79 (s, 3H), 3.14–3.09 (m, 1H), 2.23–2.17 (m, 1H), 2.11–2.05 (m, 1H), 1.95–1.87 (m, 2H), 1.78–1.73 (m, 1H), 1.48 (s, 9H), 1,47 (s, 9H); ^13^C{^1^H} NMR (125 MHz, CDCl_3_) δ 174.0, 173.0, 155.3, 130.0, 81.0, 80.2, 79.2, 53.0, 52.5, 52.0, 38.0, 36.0, 32.3, 32.0, 28.31, 28.27; FT-IR (neat) ν_max_ 3281, 2921, 2853, 1801, 1747, 1697, 1674, 1537, 1451, 1368, 1294, 1252, 1213, 1154, 1060, 1029, 1005 cm^−1^; HRMS (ESI-TOF) *m*/*z* [M + Na]^+^ calcd for C_20_H_33_IN_2_O_8_Na 579.1173, found 579.1195.



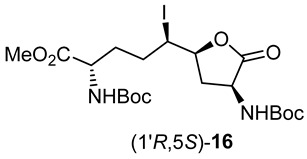



### 3.11. (1′R,5S)-3-N-(Boc)amino-5-[1′-iodo-4′-N-(Boc)amino-4′-hydroxcarbonylbutyl]-tetrahydrofuran-2-one [(1′R,5S)-***20***]

A solution of diacid **19** (1.6 g, 3.8 mmol) in acetonitrile (20 mL) was treated with Cs_2_CO_3_ (3.7 g, 11.5 mmol, 3 equiv.), stirred for 15 min, cooled to 0 °C with an ice bath, and treated with iodine (2.93 g, 11.5 mmol, 3 equiv.). The ice bath was removed. After stirring for 3–4 h, the reaction mixture had warmed to room temperature and was observed by LCMS to contain a new peak at RT = 8.1 min (C18 column, 10:90 CH_3_CN:H_2_O) with a molecular ion [M + Na]^+^ *m*/*z* 565. The reaction mixture was filtered through a pad of Celite™ and the filter cake was washed with acetonitrile (3 × 30 mL). The filtrate and washings were combined and evaporated under reduced pressure. The residue was partitioned between H_2_O (50 mL) and EtOAc (25 mL). The aqueous phase was acidified with 1 N HCl to pH 3 and extracted with ethyl acetate (3 × 50 mL). The organic extractions were combined, dried with Na_2_SO_4_, filtered, and concentrated under vacuum to afford tetrahydrofuran-2-one (1′*R*,5*S*)-**20** (2.1 g) as a pale-yellow solid, which was used without further purification.



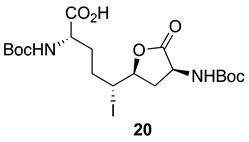



## 4. Conclusions

The copper catalyzed S_N_2′ addition of zincate derived from methyl β-iodo alaninate onto (*E*)-1,3-dichloroprop-1-ene has given useful entry into a set of protected ∆^4^-2,8-diaminoazelates (e.g., **11a**–**c**). Attempts to fold the latter linear precursors into bicyclic 5- and 7-substituted indolizidin-2-one amino acid (I^2^aa) derivatives have, however, demonstrated the challenges of achieving diastereomeric and regioisomeric selectivity in the facial differentiation of the olefin. Epoxidation and dihydroxylation were unselective and gave oxiranes **12** and hydroxy lactone **17** as inseparable mixtures of diastereomers, which were shown by LCMS analyses to be convertible into mixtures of up to four hydroxy indolizidine-2-one isomers due in part to the inability to control the intramolecular cyclization of the respective nitrogen. Moreover, diastereomeric stereochemical integrity may have also been lost due to cyclization by way of planar S_N_1 intermediates.

Iodolactonization of ∆^4^-2,8-diaminoazelic diacid **19** occurred with high facial selectivity to provide tetrahydrofuran-2-one (1′*R*,5*S*)-**16** as a single isomer. Both nitrogen of iodide **16** reacted in intramolecular S_N_2 displacements to respectively provide hydroxyproline and proline intermediates. Lactam formation provided 5- and 7-hydroxy indolizidin-2-one amino esters (3*S*,5*S*,6*S*,9*S*)-**8** and (3*S*,6*S*,7*S*,9*S*)-**9** in six steps and 21% and 17% respective overall yields from ∆^4^-2,8-diaminoazelate **11a**. Saponification of the esters (3*S*,5*S*,6*S*,9*S*)-**8** and (3*S*,6*S*,7*S*,9*S*)-**9** delivered the corresponding carboxylic acids, which are suitable for peptide synthesis.

The configuration of 5- and 7-hydroxy indolizidin-2-one amino esters (3*S*,5*S*,6*S*,9*S*)-**8** and (3*S*,6*S*,7*S*,9*S*)-**9** was assigned using a series of NMR experiments. Furthermore, X-ray analysis of ester (3*S*,6*S*,7*S*,9*S*)-**3** demonstrated that the backbone geometry within the 7-hydroxy indolizidine-2-one framework replicated that of the parent I^2^aa ester (6*S*)-**21** and mimicked the dihedral angles of the central dipeptide in a type II’ β-turn. The utility of 5- and 7-hydroxy indolizidin-2-one amino acids (3*S*,5*S*,6*S*,9*S*)-**2** and (3*S*,6*S*,7*S*,9*S*)-**3** is currently being investigated inside biologically relevant peptides and will be reported in due time.

## Data Availability

CCDC 2125339 contains the supplementary crystallographic data for this paper. These data can be obtained free of charge via www.ccdc.cam.ac.uk/data_request/cif (accessed on 20 December 2021), or by emailing data_request@ccdc.cam.ac.uk (accessed on 20 December 2021), or by contacting The Cambridge Crystallographic Data Centre, 12 Union Road, Cambridge CB2 1EZ, UK; fax: +44-1223-336033.

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
