# Peer review of "Constrained Dipeptide Surrogates: 5- and 7-Hydroxy Indolizidin-2-one Amino Acid Synthesis from Iodolactonization of Dehydro-2,8-diamino Azelates"

_molecules, 2021, doi:10.3390/molecules27010067_

Round 1
Reviewer 1 Report
To be frank, I am not sure of the immense interest of the targeted molecules. However, this is a well writen, well presented paper, and synthetic strategies are well described.
A period is lacking on line 83.
Regarding the assignment of the stereochemistry of compounds 11, is it really impossible to determine the coupling constant between the 2 H's of the double bond? In the experimental part, all the protons of these compounds are presented as a multiplet. In view of the spectra in the supp. mat., it seems to me that some coupling constants could (should) be reported. Which would still be simpler and more direct than the NOESY study. This is indeed general in NMR spectra, I suggest that authors make a little more effort to study the multiplicity of signals when possible.
In Figure 1, the representation of the two enantiomers is confusing. A careful study of the stereoselection is expected. We learn later that there is indeed no selectivity. So, why not use, in Scheme 1, the simpler drawing of Scheme 2?
Is the X-ray structure supposed to show the absolute configuration of 9? It is not always easy in the absence of a heavy atom. Did the authors use the known configuration of the starting material 10 to derive the shown absolute configuration? Or was the X-ray analysis precise enough to guarantee it? I suggest that the authors add a sentence on this subject.
Author Response
To be frank, I am not sure of the immense interest of the targeted molecules. However, this is a well writen, well presented paper, and synthetic strategies are well described.
Reply: Thank you.
A period is lacking on line 83.
Reply: the period has been added.
Regarding the assignment of the stereochemistry of compounds 11, is it really impossible to determine the coupling constant between the 2 H's of the double bond? In the experimental part, all the protons of these compounds are presented as a multiplet. In view of the spectra in the supp. mat., it seems to me that some coupling constants could (should) be reported. Which would still be simpler and more direct than the NOESY study. This is indeed general in NMR spectra, I suggest that authors make a little more effort to study the multiplicity of signals when possible.
Reply: As requested, the coupling constants (15 Hz) were measured and are now discussed in the manuscript. Although the values suggest E-isomer, without the other isomer, there is no point of comparison. The NOSEY spectra provide confirmation of the E isomer without need of the Z-olefin.
In Figure 1, the representation of the two enantiomers is confusing. A careful study of the stereoselection is expected. We learn later that there is indeed no selectivity. So, why not use, in Scheme 1, the simpler drawing of Scheme 2?
Reply: In Figure 1, Scheme 1 and Scheme 2, diastereomers are being represented. In Figure 1, diastereomers of different indilizidinone amino acids with and without substituents are represented. In Scheme 1, the two possible epoxide diastereomers from m-CPBA oxidation are represented. In Scheme 2, the same two are shown without specific designation of the 4R,5R or 4S,5S isomers. Although the two images in Scheme 1 for the 4R,5R and 4S,5S isomers may be redrawn, this will not improve clarity.
Is the X-ray structure supposed to show the absolute configuration of 9? It is not always easy in the absence of a heavy atom. Did the authors use the known configuration of the starting material 10 to derive the shown absolute configuration? Or was the X-ray analysis precise enough to guarantee it? I suggest that the authors add a sentence on this subject.
Reply, as per the Reviewer's comments, without a heavy atom, the absolute stereochemistry is difficult to assess. Instead, the relative stereochemistry is assigned based on the starting L-amino acids and the unlikely event of epimerization of these chiral centers during the synthesis of indolizidinone 9. The text was modified accordingly: "The relative configurational assignments for 7-hydroxy Boc-I2aa-OMe (6S,7S)-9 were confirmed by X-ray analysis of crystals grown from a dichloromethane in hexanes mixture (Figure 3)."
Reviewer 2 Report
In this manuscript, the authors conveyed the constrained dipeptide surrogates 5- and 7-hydroxy indolizidin-2-one N-(Boc)amino acids from L-serine via iodolactonization. Moreover, these 5- and 7-hydroxy indolizidin-2-one amino acid cores might be medicinally important and biologically worthy. Overall, the scholarly quality of this paper is not satisfactory for acceptance in molecules in the present form. However, there are some further remarks that requisite to be clarify before its publishing.
My comments, suggestions, and questions to authors:
- Manuscript contains potentially interesting data but it is not well organized and contains some severances and the numbering of
formulas in the Schemes are confused. Why compound 8 and 9 appears just after the compound 14? - Some places they included scheme/figure numbers and other places not see line 34, 94, 129, 150, 185, 208, etc. and that need to be fixed in all cases.
- In all schemes time and temp. should be included.
- In Scheme?? Line 132/138 <<diaminoazelate>> is it compound 10a??
- Line 159-161 in page 6: the structure of compound 19 has not been shown anywhere in MS and 159-161 should be rephrase.
- The manuscript is riddled with typos and mistakes and should be thoroughly checked.
- Traces of impurities found in some of the spectrums and NMR chemical shift ranges should be 0-10 ppm (1H), 0-200 ppm (13C NMR) in all cases and peak height should be reasonable and suggest authors to recheck data.
Author Response
In this manuscript, the authors conveyed the constrained dipeptide surrogates 5- and 7-hydroxy indolizidin-2-one N-(Boc)amino acids from L-serine via iodolactonization. Moreover, these 5- and 7-hydroxy indolizidin-2-one amino acid cores might be medicinally important and biologically worthy. Overall, the scholarly quality of this paper is not satisfactory for acceptance in molecules in the present form. However, there are some further remarks that requisite to be clarify before its publishing.
Reply: as requested, the comments are addressed below.
My comments, suggestions, and questions to authors:
- Manuscript contains potentially interesting data but it is not well organized and contains some severances and the numbering of
formulas in the Schemes are confused. Why compound 8 and 9 appears just after the compound 14?
Reply: as requested, compounds 8 and 9 have been added to Figure 1 to facilitate the readers understanding.
- Some places they included scheme/figure numbers and other places not see line 34, 94, 129, 150, 185, 208, etc. and that need to be fixed in all cases.
Reply: unfortunately, the texts mentioned were removed when the manuscript was formatted to the journal style and were not checked before submission to the Reviewers. As requested, these omissions and the numbering of the images have been rectified.
- In all schemes time and temp. should be included.
Reply: as requested, these details have now been added.
- In Scheme?? Line 132/138 <<diaminoazelate>> is it compound 10a??
Reply: the numbering has been corrected.
- Line 159-161 in page 6: the structure of compound 19 has not been shown anywhere in MS and 159-161 should be rephrase.
Reply, Scheme 4 has been corrected to show 19. The numbering has been corrected.
- The manuscript is riddled with typos and mistakes and should be thoroughly checked.
The text has been revised carefully.
- Traces of impurities found in some of the spectrums and NMR chemical shift ranges should be 0-10 ppm (1H), 0-200 ppm (13C NMR) in all cases and peak height should be reasonable and suggest authors to recheck data.
Reply, as requested, trace impurities were minimized and full chemical shift ranges were employed in the revised spectra.
Round 2
Reviewer 2 Report
The manuscript has been improved by authors by addressing all the comments/suggestion from the reviewers. Therefore, the manuscript is recommended for publication in this form.